# Electrospun Silicon Dioxide/poly(vinylidene fluoride) Nanofibrous Membrane Comprising a Skin Multicore–Shell Nanostructure as a New High-Heat-Resistant Separator for Lithium-Ion Polymer Batteries

**DOI:** 10.3390/polym16131810

**Published:** 2024-06-26

**Authors:** Young-Gon Kim, Bo Gyeong Jeong, Bum Jin Park, Heejin Kim, Min Wook Lee, Seong Mu Jo

**Affiliations:** 1Institute of Advanced Composite Materials, Korea Institute of Science and Technology, Chudong-ro, Bongdong-eup, Wanju-gun 55324, Republic of Korea; 2Nanomaterials Science and Engineering, Korea University of Science and Technology, Gajeong-ro, Yuseong-gu, Daejeon 34113, Republic of Korea

**Keywords:** SiO_2_/PVdF blend fiber, electrospinning, skin multicore–shell, separator, high thermal property, lithium secondary polymer battery

## Abstract

Porous silicon dioxide (SiO_2_)/poly(vinylidene fluoride) (PVdF), SiO_2_/PVdF, and fibrous composite membranes were prepared by electrospinning a blend solution of a SiO_2_ sol–gel/PVdF. The nanofibers of the SiO_2_/PVdF (3/7 wt. ratio) blend comprised skin and nanofibrillar structures which were obtained from the SiO_2_ component. The thickness of the SiO_2_ skin layer comprising a thin skin layer could be readily tuned depending on the weight proportions of SiO_2_ and PVdF. The composite membrane exhibited a low thermal shrinkage of ~3% for 2 h at 200 °C. In the prototype cell comprising the composite membrane, the alternating current impedance increased rapidly at ~225 °C, and the open-circuit voltage steeply decreased at ~170 °C, almost becoming 0 V at ~180 °C. After being exposed at temperatures of >270 °C, its three-dimensional network structure was maintained without the closure of the pore structure by a melt-down of the membrane.

## 1. Introduction

Lithium secondary batteries have been recently employed in portable power applications, and they are particularly very appropriate for larger power applications, such as powering electric vehicles (EVs) and ensuring large capacities for energy storage devices, owing to their high energy capacities and relatively low weights. However, these batteries (for EVs or other applications), which require substantial amounts of energy, must overcome several safety issues, such as physical, electrical, and thermal abuses. In particular, the thermal runaway process proceeds in three stages. The process involves the breakdown of the solid electrolyte interface (SEI) layer, which causes electrolyte reduction at the exposed lithiated graphite anode around 90–120 °C, the reaction initiated at the active cathode around 140–160 °C, and the high-rate thermal runaway at ≥180 °C due to oxygen generation from cathode decomposition, followed by electrolyte oxidation [1,2,3]. The heat generation by commercially available small cells during charging and discharging does not present significant concerns, as the small size facilitates heat dissipation. However, the above safety issues could be more significant in larger cells. Notably, a larger cell requires a more dimensionally stable (high-temperature melt integrity) separator because of possible short-circuiting and the generation of large amounts of heat. Currently, commercially available polyolefin-based separators exhibit thermal shut-down functions at ~130 °C owing to the melting of polyethylene (PE); thus, these separators are unstable at higher temperatures [4].

Some studies have found new separators that are composed of coating thermally stable materials to enhance thermal stability. To enhance thermal stability, several studies have developed novel separators or polymer electrolytes comprising modified polymer structures by applying hybrid electrolytes, crosslinking or grafting, coating with thermally stable materials, and fabricating organic–inorganic composite membranes. Commercially available polypropylene (PP)-based microporous separators are laminated with nonwoven materials under heat and pressure. However, these products can offer dimensional stability up to only 167 °C (the melting point of PP), and their ion-transport channels are generally blocked during lamination, greatly reducing the cell efficiency. Recently, a new commercial high-heat-resistant separator was prepared by the ceramic coating of a commercially available polyolefin-based separator with a higher porosity (~60%) than the conventional porosity (40%) to avoid porosity reduction during the ceramic coating [5,6,7,8,9,10,11,12]. Although this type of separator exhibits good thermal properties, demonstrating a small thermal shrinkage of ~10% at 150 °C, the resulting cell displays lower power performance (up to 10%) than that comprising the uncoated PE separator. Further, electrospinning represents an efficient fabrication method for porous membranes comprising continuous ultrafine fibers with diameters ranging from a few micrometers to a few nanometers [13,14]. As-electrospun membranes exhibit high porosity and offer very good ion transfer channels, ensuring good ionic conductivity and power performance, they are good candidates for lithium-ion battery (LIB) separators [15,16,17,18,19].

In this study, we fabricated skin-layered multicore–shell silicon dioxide (SiO_2)_/poly(vinylidene fluoride) (PVdF), SiO_2_/PVdF, and nanofibers from a two-phase blend solution by the electrospinning technique. The rapid solvent evaporation during electrospinning generated an abnormal SiO_2_ skin layer and a nanofibril of the electrospun fiber. As this type of membrane ensures much better temperature tolerance than the nanofibrous membranes obtained from an electrospun fumed silica nanoparticle/PVdF mixture, we investigated the suitable power performance required for the application of the resulting cell in electric vehicles (EVs) or hybrid EVs.

## 2. Experimental Section

### 2.1. Materials

The chemicals were supplied by Simga-Aldrich, St. Louis, MO, USA and were used as received without further purification, including tetraethoxysilane (TEOS), ethyl alcohol, N, N-dimethylformamide (DMF), and fumed silica whose particle sizes are chain-like aggregates a few tenths of a micron. Poly(vinylidene fluoride) (PVdF, Kynar®761) was supplied by Atofina, Paris, France.

### 2.2. Electrospun SiO_2_/PVdF Immiscible Blend-Based Composite Membrane

To obtain a sol–gel solution via the hydrolysis and condensation of TEOS, a TEOS:ethyl alcohol:H_2_O:HCl (1:2:2:0.01 mole ratio) solution was first heated to 75 °C to initiate a sol–gel reaction for ~2 h. To prevent the continuation of the sol–gel reaction, the resulting viscose solution was diluted by adding DMF. Thereafter, a silica sol–gel solution was mixed with a PVdF solution in DMF, followed by stirring at 60 °C for 1 h. The solid content of the silica sol–gel/PVdF (3/7 wt. ratio) blend solution was 21 wt.%. 

Several immiscible silica sol–gel/PVdF (1/9, 3/7, and 5/5 wt. ratio) blend solutions were electrospun to form SiO_2_/PVdF blend fiber-based composite membranes at an applied voltage of 10.5 kV, a tip-to-grounded collector distance of 14 cm, a needle size of 30 G, and a solution feeding rate of 30 μL/min. 

To control the porosity and improve the mechanical properties, postprocesses, including heat-pressing and stretching at 120 °C and 150 °C, respectively, were performed using the hand-made roll press with rod-type heaters and stretching apparatus in the heating chamber.

The morphology of the electrospun fibers was characterized by scanning electron microscopy (SEM; S4200, Hitachi, Tokyo, Japan) and the cross-sectional images and elemental-line mappings of the fibers were observed by transmission electron microscopy (TEM, Titan and Tecnai G2, FEI, Oregon, US/CM30, Philips, Amsterdam, The Netherlands) equipped with energy-dispersive X-ray spectroscopy (EDX).

The porosity of the composite membrane was determined by the n-butanol (BuOH) uptake method. The uptake of the electrolyte solution was determined by soaking the fibrous membranes (size, 3 cm × 3 cm) in a 1 M LiPF_6_–ethylene carbonate (EC)/propylene carbonate (PC)/diethyl carbonate (DEC)/VC (30.8/15.0/39.2/2.0 wt.%) solution [20].

The mechanical properties were determined by a universal testing machine (Instron, Model 4464) according to the American Society for Testing and Materials standard D882-95a [21]. Additionally, the composite membranes were subjected to thermal shrinkage tests through storage at 200~300 °C for 2 h.

### 2.3. Preparation of the Electrospun Polymer Electrolyte Membrane (EPEM) and a Proto-Type Cell 

The EPEM was prepared by injecting a specific amount of LiPF_6_–EC/PC/DEC/VC (30.8/15.0/39.2/2.0 wt.%) into the electrospun SiO_2_/PVdF composite membrane and removing the excess electrolyte solution in a glove box filled with argon gas (H_2_O < 1 ppm) at 25–27 °C. The prototype cell was prepared using EPEM, a mesocarbon microbead (MCMB) anode, and a lithium cobalt oxide (LiCoO_2_) cathode. Thereafter, the cell (3 cm × 4 cm) was vacuum sealed in an aluminum-plastic pouch with a capacity of 21.1 mAh (theoretical capacity, 141 mAh/g).

### 2.4. Measurements of the Thermal Stabilities of the EPEM and Prototype Cell 

To investigate the thermal stability of EPEM, the electrochemical impedances were measured using a prototype cell (LiCoO_2_//EPEM//MCMB) and EPEM sandwiched between two blocking stainless steel (SUS) electrodes (SUS//EPEM//SUS).

The ionic conductivities of EPEM were measured by the alternating current (AC) impedance method using electrochemical impedance spectroscopy (IM6e, Zahner-elektrik, Kronach, Germany) over a frequency range of 100 mHz–1 MHz at an amplitude of 10 mV. The SUS//EPEM//SUS cell was heated at a rate of 5 °C/min from 40 °C to 300 °C. After reaching each desired temperature, a 5 min rest period was applied before recording the impedances.

To investigate the thermal stability of the prototype cell, a multichannel potentiostat equipped with impedance modules (BioLogic, Orlendo, FL, USA) was used to measure the electrochemical impedance at 1 kHz in the frequency range of 100 mHz–1 MHz, with an AC voltage of 10 mV. The prototype cell was charged to a 50% state of charge (SOC) and heated at a rate of 5 °C/min from 40 °C to 300 °C to investigate its thermal stabilities. The rest time was 5 min per 10 °C, and the impedance was measured along with the open-circuit voltage (OCV) for 10 s.

### 2.5. Battery Performances 

The hybrid pulse power characterization (HPPC) test for investigating the power performance of the cell was typically conducted based on the partnership for a new generation of vehicle (PNGV) battery test manual [22,23], as shown in Appendix A. This HPPC test was repeated using a battery cycler. The discharge- and regenerated-pulse power capabilities were calculated at each depth-of-discharge (DOD) increment from the OCV and resistance, showing the V_MIN_ (cell discharge power capability) and V_MAX_ (cell regenerated power capability), as displayed in Equations (1) and (2), respectively.
Discharge-pulse power capability = *V_MIN_* · (*OCV_discharge_* − *V_MIN_*)/*R_discharge_*(1)
Regenerated-pulse power capability = *V_MAX_* · (*V_MAX_* − *OCV_regen_*)/*R_regen._*(2)

The commercial PE separator with a porosity of ~60% (SKC provided, Seoul, Republic of Korea) was used as the reference material for evaluating the battery performance. 

The temperature dependences of the discharge capacity of the prototype cells were investigated using a battery cycler (WBCS3000, WonATech Co., Seoul, Repulick of Korea). The cells were cycled 5 times at a 0.2C rate to complete the formation of the SEI. The cells were charged at C/10, after which they were discharged with cut-off voltages of 2.75 and 4.2 V at a 1C rate and various temperatures, from 0 to 60 °C, at 20 °C intervals. 

## 3. Results and Discussion

The SEM images of the internal and external morphologies of the electrospun SiO_2_/PVdF membrane are shown in Figure 1. Figure 1a shows the SEM images of the electrospun SiO_2_/PVdF blend nanofiber-based composite membrane. The electrospun SiO_2_/PVdF blend nanofiber exhibited a smooth surface and proper interfiber bonding, resulting in a stable, thin-membrane structure. The thickness of the as-electrospun fibrous membrane was ~85 μm. The morphology of the electrospun membrane greatly depended on the electrospinning conditions, such as the polymer concentration, solution viscosity, nozzle-to-collector distance, and humidity. The deposition of completely dried or coagulated fibers on the collector yielded a membrane with enhanced porosity and very weak interfiber bonding. However, the control of solvent evaporation from the polymer jet during electrospinning yielded a stable, thin-membrane structure. The as-electrospun SiO_2_/PVdF membrane exhibited a porous structure (porosity, 86–89%), with a three-dimensional network comprising the SiO_2_/PVdF blend fibers with an average diameter of ~310 nm and a narrow fiber-diameter distribution. As shown in Figure 1b, the stretched membrane revealed the presence of a thin skin layer and core bulk inside the SiO_2_/PVdF fiber. The cross-section of the electrospun fiber (Figure 1c) correlated with the result obtained from the physically stretched fiber. The extraction of PVdF from the SiO_2_–/PVdF fibers using acetone for seven days indicated that SiO_2_ produced the thin skin layer and fiber-like core bulk in the as-spun SiO_2_/PVdF fiber.

TEM analysis was carried out using specimens prepared by the cryo-ultramicrotome method in order to establish the complex internal morphology of SiO_2_/PVdF fiber. Figure 2a shows the high-angle angular dark field (HAADF) image of the SiO_2_/PVdF fiber cross-section. The SiO_2_/PVdF fiber comprised a thin skin layer, a core bulk, and a few spots inside the bulk. Figure 2b,c show the elemental mapping images of the SiO_2_/PVdF fiber. Although the carbon mapping of the core bulk did not clearly clarify the elemental distribution of the exact point owing to the inherently restricted resolution of the scanning TEM mode, we concluded that a clear hollow was prominent and corresponded to SiO_2_. These revealed the elemental composition of the SiO_2_/PVdF fiber in which the thin skin layer and nanofibrils inside the core bulk comprised SiO_2_- and PVdF-occupied core bulks. This unique morphology, a skin-layered multicore–shell fiber, was named. Moreover, the dark domains inside the bulk were grown along the direction of the elongation of the electrospun fibers (Figure 3).

Figure 4a shows the SEM images recorded after the calcination of an as-electrospun composite membrane at 700 °C. The electrospun pure PVdF nanofiber generally changed into granular-type carbon through high-temperature carbonization [24]. As shown in Figure 4a, the skin and nanofibrillar structures, which were assumed to be the SiO_2_ component, were observed after calcination at 700 °C, indicating that the electrospun SiO_2_/PVdF blend nanofibers comprised a SiO_2_-based skin structure and a nanofibrillar structure. The nanostructures, including the skin multicore–shell nanostructure, the mechanism of their formation, and the thermal stabilities of the electrospun SiO_2_/PVdF composite membranes with several blend ratios of SiO_2_/PVdF, were discussed in other papers [25,26]. Additionally, the thickness of SiO_2_ comprising the thin skin layer could be readily tunable depending on the weight proportion of SiO_2_ and PVdF.

Figure 5a–c show the cross-sectional TEM images of the electrospun fibers comprising 1:9, 3:7, and 5:5 weight ratios of SiO_2_ and PVdF. The schematics are illustrated in Figure 5d. It is noteworthy that this trend was consistent with the occasions in the SiO_2_-PAN blend system. Appendix A demonstrate the internal morphologies of SiO_2_-PAN = 5:5 and 6:4. There were not only identical multicore–shell morphologies but also facilely modulated SiO_2_ skin layer as shown in the SiO_2_-PVdF blend system. The EDS line profile along the marked red line in Appendix A re-emphasized that carbon and silicon were read by turns and this electrospun fiber had skin-layered multicore–shell morphology.

It is crucial to comprehend the mechanism of the formation of the skin-layered multicore–shell morphology of the SiO_2_/PVdF fibers to further modify and effectively utilize the system for diverse applications. The internal morphology of electrospun fumed silica–PVdF composite nanofiber was observed by SEM and TEM (Appendix A). No skin-layered morphology was observed in electrospun fumed silica–PVdF composite; rather it appeared like poorly dispersed conventional composite fibers as previously reported [18]. This is because most groups have prepared electrospun SiO_2_/PVdF composite fibers by adding precursors or particles at the initial stage of solution preparation.

Regarding the identity of the prepared SiO_2_/PVdF nanofibers, most of the TEOS sol–gel reaction proceeded before the mixing of SiO_2_ and PVdF, and we assumed that SiO_2_ grew as an inorganic polymer chain, following the sol–gel reaction. Once the SiO_2_ sol–gel and PVdF solutions were mixed, the SiO_2_/PVdF blend solution might form an emulsion phase rather than a homogeneous phase, as there were still large numbers of residual hydroxyl (–OH) groups on the Si domain of the huge mass of the SiO_2_ complex [27]. However, an excess amount of DMF in the solution might countervail the activity of the –OH groups on the Si domain and make it transparent. Moreover, from the extant study on the preparation of polymer nanotubes via electrospinning by Li et al. [28], TEOS was soluble in ethanol, and when the blend solution was subjected to an electric field, TEOS exhibited a concentration gradient along with the ethanol evaporation.

Next, we proposed the possible mechanism for the formation of the skin-layered multicore–shell morphology of the SiO_2_/PVdF nanofiber. A silica sol–gel/PVdF immiscible blend solution for the electrospinning was a mixed solution of the two immiscible polymers, although the blend solution appeared like a transparent solution. The phase separation between the silica-rich and PVdF-rich phases was observed after the prolonged storage of the mixed solution. In the very first electrospinning step governed by the Taylor cone, the SiO_2_/PVdF emulsion solely underwent a dragging force along the tip-to-collector direction. A substantial evaporation degree of the solvent near the jet surface was initiated, and the SiO_2_ domains were enlarged in the ohmic flow zone. The whipping motion of the electrospinning jet accelerated vigorous fiber elongation and continuous solvent evaporation. However, as ethanol exhibits relatively higher vapor pressure than DMF under identical conditions, it evaporated more rapidly than that on the emulsion surface. Simultaneously, the SiO_2_ phase diffused out near the emulsion surface with ethanol, after which SiO_2_ generated a skin layer on the SiO_2_/PVdF fiber surface owing to the evaporation of the solvents. Conversely, the core bulk maintained the emulsion phase and obtained a directional force, developing an aligned SiO_2_ nanofibrous morphology and a PVdF bulk core. The procedures for developing the skin-layered multicore–shell morphology during electrospinning are schematically illustrated in Figure 6.

The unique morphology of the skin-layered multicore–shell fiber bestowed superior thermal stabilities on the SiO_2_/PVdF membranes. These excellent features were authenticated by further morphological and thermal analyses. The thermal degradation characteristics of the membranes are shown in Appendix A. The dominant weight loss of all membranes by heat was observed at 486 °C and was due to the decomposition of the PVdF fragments exhibiting relatively low thermal stability compared with SiO_2_. A few degradations at <486 °C might be attributed to impurities, such as the –OH groups on the Si domain and solvents. The thermal shrinkage test was conducted using the same composition of the SiO_2_/PVdF and fumed silica–PVdF membranes. The changes in the dimensions of the SiO_2_/PVdF membranes at different temperatures are depicted in Figure 7. The electrospun SiO_2_/PVdF blend membrane only exhibited slight deformation (<3%) at 200 °C for 2 h. Furthermore, the SiO_2_/PVdF membrane retained 84% of its original shape at 300 °C for 2 h. Conversely, the fumed silica–PVdF membrane could not withstand the temperature of 200 °C and even shriveled up at 300 °C because of the low melting temperature of PVdF (~165 °C). These excellent thermal properties might be attributed to the fact that the multicore of the SiO_2_ nanofibril acted as a rigid backbone, whereas the skin layer preserved the melting of the polymer. Consequently, the skin-layered multicore–shell-structured SiO_2_/PVdF membrane simultaneously exhibited unique thermal stabilities and polymer-like flexibility.

The pore structure of the electrospun SiO_2_/PVdF blend nanofiber membrane was maintained after the postprocesses, including heat-pressing at 120 °C, followed by heat-stretching at 150 °C. The porosity of the composite membrane was controlled to ~60% through the post-treatment, including heat-pressing and heat-stretching. Regarding the as-electrospun pure PVdF membrane, hot-pressing at >120 °C yielded a film-like structure with a collapsed pore structure owing to the partial melting of PVdF (m.p. 165 °C) under these conditions. Heat-pressing at extremely low temperatures (<120 °C) did not increase the interfiber bonding, although it retained the pore structure. Therefore, these membrane types could not be extensively stretched, even at high temperatures, because of their extremely weak interfiber bonding, resulting in poor mechanical properties

Figure 8 shows the SEM and photo images of the composite membranes after (a) heat-pressing at 120 °C, followed by heat-stretching at 150 °C. The as-electrospun SiO_2_/PVdF composite membrane was slightly oriented along the machine direction after 71.4% heat-stretching, whereas the fibers were randomly distributed before heat-stretching. The composite membrane exhibited strong interfiber bonds after hot-pressing because of the partial melting of PVdF at the intersecting points. However, the membrane retained its original pore structure and good dimensional stability after the postprocesses. The strong interfiber bonding in the electrospun fibrous membrane helped improve the orientation of the molecular chain of the polymer along the fiber axis through further heat-stretching, indicating high mechanical strength. Therefore, the high orientation of the molecular chain after the postprocesses improved the mechanical properties of the composite membrane. Table 1 demonstrates the increase in the mechanical properties of the electrospun SiO_2_/PVdF blend membrane through heat-pressing and heat-stretching. The heat-pressed composite membrane was broken under 84.2% heat-stretching, which was a very low heat-stretching ratio compared with that of the electrospun PVdF membrane (heat-stretching ratio, 500%–600%). The white-circled part in Figure 8b also shows the broken shape of the skin layer of the electrospun SiO_2_/PVdF composite nanofiber structure due to heat-stretching.

The skin multicore–shell of the SiO_2_ component is thought to be responsible for a lower heat-stretching ratio than that of pure PVdF nanofiber. However, the composite membrane, which was heat-pressed and heat-stretched, showed a low thermal shrinkage of approximately below 3% after a heat treatment at 200 °C for 2 h. Low thermal shrinkage of the composite membrane (high thermal stability) is thought to be due to the SiO_2_-based skin and nanofibrillar structure of the electrospun SiO_2_/PVdF blend nanofiber. Therefore, we anticipate that it can be a superb solution in high-temperature endurable applications such as separators for lithium batteries, water treatment, and flexible thermal insulation.

Figure 9 shows the ionic resistivity of the conductivity cells (SUS//EPEM//SUS), in which EPEM was sandwiched between the two blocking stainless steel electrodes, as a function of the temperature. Regarding the PE separator as a reference, the ionic resistivity increased rapidly at ~120 °C owing to the melting of the PE separator, which supported the thermal shut-down behavior of the PE separator. However, the ionic resistance of the conductivity cells containing EPEM did not increase significantly. The conductivity cell did not display the shut-down behavior due to the complete closure of the membrane pore structure by the melting of PVdF (m.p = 165 °C) or swollen PVdF up to a temperature of 300 °C, indicating high thermal stability. In a previous study, the differential scanning calorimetry curve of the PVdF membrane containing an electrolyte solution displayed two major peaks, which corresponded to the melting of the swollen and partially swollen polymers on the surface and inner region of the fibers, respectively [18,29]. The SiO_2_-based skin structure in the composite nanofiber may prevent the closure of the pore structure by the melt flow of the swollen parts of PVdF or PVdF itself.

Figure 10 shows the thermal stability of the prototype cell (MCMB anode/EPEM/LiCoO_2_ cathode). The OCV and impedance at 1 kHz were measured in the temperature range of 40 °C–300 °C to investigate the thermal stability of the electrospun SiO_2_/PVdF composite membrane in the lithium cell containing LiCoO_2_ and MCMB as the cathode and anode, respectively. The cell was charged to 50% SOC. Regarding the prototype cell using the PE separator (MCMB anode/PE/LiCoO_2_ cathode), the OCV gradually decreased at 120 °C and was almost zero at 140 °C. Additionally, the impedance at 1 KHz sharply increased at 130 °C, similar to the sharp increase in the ionic resistivity displayed in Appendix A. This is a typical shut-down behavior due to the melting of the PE separator at ~130 °C. The shut-down behavior of the commercial PE separator was observed regardless of the presence of the electrode materials, as the electrode materials started decomposing at a much higher temperature than the melting point of PE [30,31]. In the prototype cell comprising the composite membrane, the AC impedance started increasing sharply at ~225 °C, dissimilar to the ionic resistance behavior of the conductivity cell, as shown in Figure 9; the OCV decreased steeply at ~170 °C, almost reaching 0 V at ~180 °C. The inset SEM image in Figure 10 also shows the composite membrane after the OCV and AC impedance measurements of the prototype cell up to 270 °C (the dashed square in Figure 10). Although this composite membrane was exposed to temperatures of >270 °C, its three-dimensional network structure was maintained without the closure of the pore structure. LiCoO_2_ in LIBs thermally decomposes through O_2_ release at ~250 °C, and the electrolyte solution degrades on the electrode surface [2]. These reactions could have caused the rapid increase in the AC impedance of the prototype cell containing the SiO_2_/PVdF composite membrane, dissimilar to the conductivity cell without the electrode materials. A sufficient increase in the ionic resistance of the thermal shut-down function might require the complete closure of the pore structure through the melt flow of PVdF exhibiting a high melt viscosity. During the thermal stability test of the prototype cell, the swollen PVdF or PVdF itself of the composite membrane might have been overheated by the large amount of heat generated from the thermal decomposition of the electrodes, significantly reducing the viscosity. The pore structure of the composite membrane might be sufficiently broken by the overheated PVdF in the swollen composite membrane of the prototype cell. However, the SiO_2_-based skin structure in the electrospun SiO_2_/PVdF blend nanofiber might prevent the closure of the membrane pore structure from a flow of the overheated PVdF melt (Figure 9). Therefore, the unstable interface formed by electrode and electrolyte decomposition might further account for the sharp increase in the AC impedance and steep decrease in OCV rather than the pore structure closure through PVdF melting. These results confirmed that the electrospun SiO_2_/PVdF composite membrane exhibited high thermal stability in the LIB at temperatures of >200 °C without the melting of the separator.

Conversely, the ionic resistance of the conductivity cell (Figure 9) did not increase significantly at temperatures of >300 °C, indicating high thermal stability without any thermal runaway. The thermal stability of the prototype cells comprising LiCoO_2_ and MCMB was lower than that of the conductivity cell without the electrode materials. This further indicated that the thermal stability of the prototype cell was lower because of the decomposition of the electrode and/or degradation of the electrolyte solution but not the melting of the electrospun composite membrane.

We investigated HPPC for evaluating the power performance of the composite membrane. Conventional commercial separators for improving heat resistance were laminated or coated with heat-resistant materials. However, these materials can greatly reduce the cell efficiency, such as power performance, by blocking their ion-transport channels. Power performances in larger applications, such as EVs and large-capacity energy storage devices, are crucial to the rapid charge and discharge of the cell. As shown in Figure 11, the discharge- and regenerated-pulse power capabilities of the composite membrane at 50% DOD were similar to or slightly higher than those of the PE separator with similar porosity (~60%). Power performance is known to be closely related to the porosity or air permeability of a separator because separators support the ion-transport channels. Put differently, these results indicate that the impedance of the electrospun SiO_2_/PVdF composite membrane is similar to or lower than those of conventional composite membranes during discharge and regeneration. The electrolyte-compatible SiO_2_/PVdF composite membrane might account for a lower impedance than that of a commercial PE separator with similar porosity. Therefore, we anticipate that the electrospun SiO_2_/PVdF composite membrane itself, without any lamination or coating, can be used as a high-heat-resistant polymer electrolyte membrane and function as a large battery with high power performance owing to its high porosity.

The discharge capacities of the prototype cells comprising EPEM in 1 M LiPF_6_–EC/PC/DEC/VC (30.8/15.0/39.2/2.0 wt.%) were measured in the temperature region of 0 °C and 60 °C. As shown in Figure 12a, the prototype cell with EPEM exhibited good discharge capacities in the operating temperature range. However, the discharge capacity decreased greatly, particularly in the low-temperature region (0 °C). Figure 12b shows the discharge capacities of the prototype cells with the composite membrane exhibiting a porosity of ~40%. The composite membrane with lower porosity displayed lower discharge capacity. The porosity of the composite membrane was controlled by heat-pressing the composite membrane. Heat-pressing may produce closed pore structures. An increase in the closed pore might produce poor discharge capacity owing to decreased electrolyte uptake and limited ion transfer.

## 4. Conclusions

SiO_2_/PVdF blend fiber-based composite membranes were prepared by electrospinning from a silica sol–gel solution/PVdF immiscible blend solution. The blend nanofibers comprised skin and nanofibrillar structures, which were SiO_2_ components. Additionally, the thickness of SiO_2_ comprising the thin skin layer was readily tunable depending on the weight proportions of SiO_2_ and PVdF. This nanostructure supported the low thermal shrinkage of the resulting composite membrane (~3%) at 200 °C for 2 h. Thus, this composite membrane, without any lamination or coating, can be used as a high-heat-resistant polymer electrolyte membrane and can function as a large high-power performance battery owing to its high porosity.

## Figures and Tables

**Figure 1 polymers-16-01810-f001:**
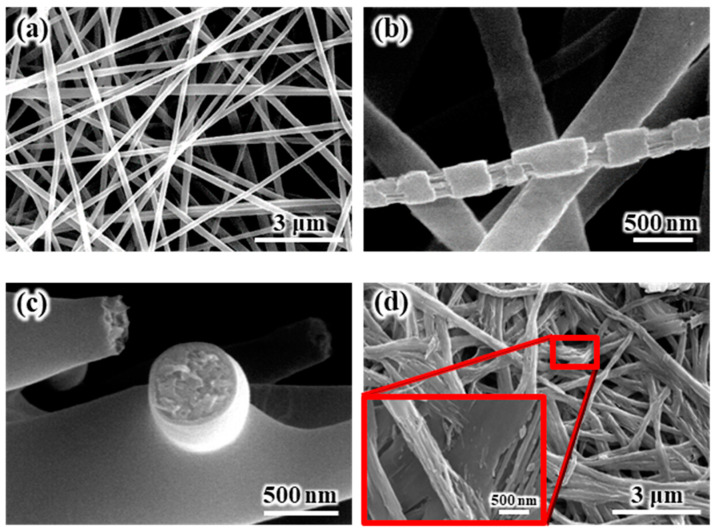
SEM images of the (**a**) as-spun, (**b**) stretched, and (**c**) cross-sectional SiO_2_/PVdF skin-layered multicore–shell electrospun membrane. (**d**) Membrane after removing the PVdF component by solvent extraction.

**Figure 2 polymers-16-01810-f002:**
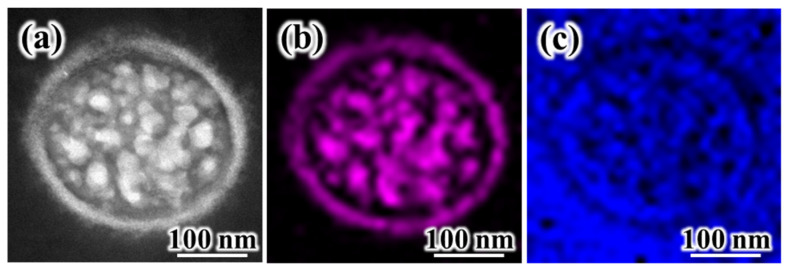
TEM analysis of the cross-section of the SiO_2_/PVdF (3:7 wt. ratio) electrospun membrane. (**a**) HAADF images; elemental mappings of (**b**) silicon (purple) and (**c**) C (blue). The specimen used in TEM analysis was prepared by the microtome method under the cryo condition.

**Figure 3 polymers-16-01810-f003:**
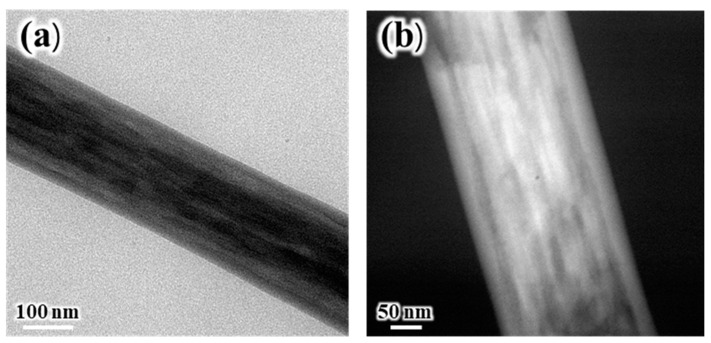
Longitudinal section of the TEM images of the SiO_2_/PVdF (5:5 wt. ratio) electrospun fiber. The specimen employed for the TEM analysis was prepared by the microtome method under the cryo condition: (**a**) bright-field image and (**b**) high-angle angular dark field (HAADF) image.

**Figure 4 polymers-16-01810-f004:**
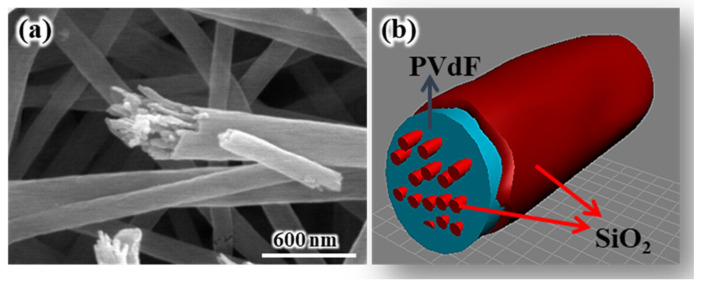
(**a**) SEM images after the calcination of the as-electrospun composite membrane at 700 °C. (**b**) Schematic of the skin multicore–shell nanostructure of the electrospun SiO_2_/PVdF composite nanofiber.

**Figure 5 polymers-16-01810-f005:**
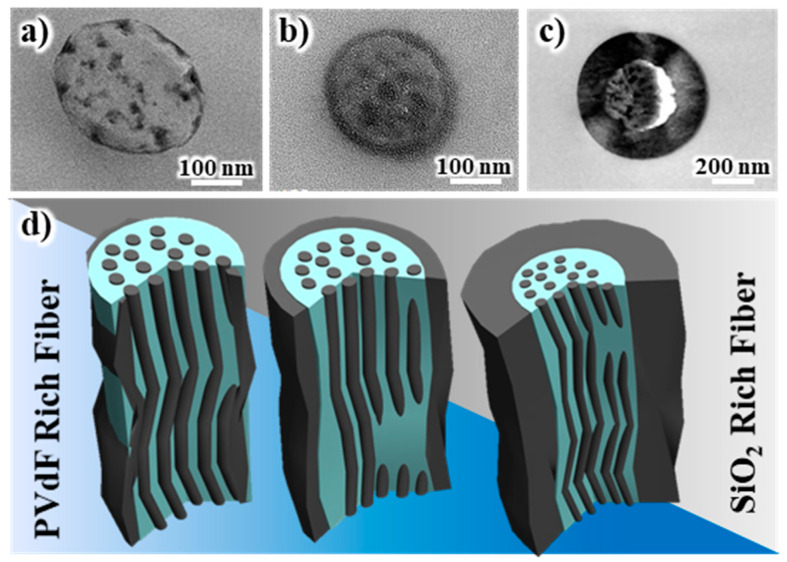
Cross-sectional TEM bright-field images of the readily tunable skin-layered multicore–shell SiO_2_/PVdF electrospun. (**a**) SiO_2_/PVdF (1:9 wt. ratio), (**b**) SiO_2_/PVdF (3:7 wt. ratio), (**c**) SiO_2_/PVdF (5:5 wt. ratio). (**d**) Schematic of those fibers, including their internal morphologies.

**Figure 6 polymers-16-01810-f006:**
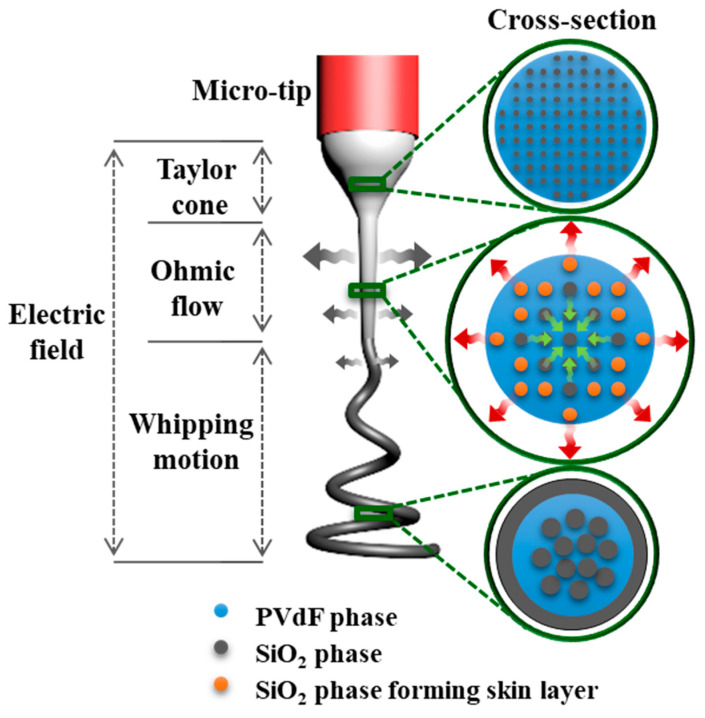
Schematic of the SiO_2_/PVdF electrospinning procedure and cross-section of each stage of the Taylor cone, Ohmic flow, and whipping motion.

**Figure 7 polymers-16-01810-f007:**
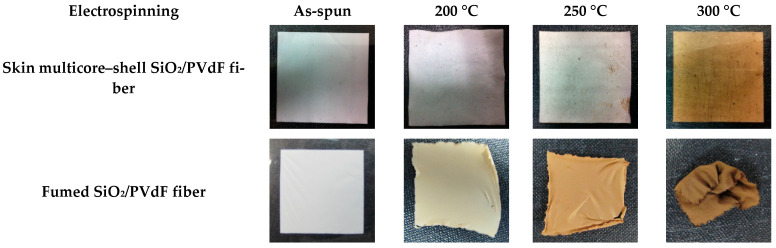
Photos of the membranes exposed to temperatures of 200 °C, 250 °C, and 300 °C for 2 h and the as-spun membrane (SiO_2_/PVdF = 3:7 wt. ratio).

**Figure 8 polymers-16-01810-f008:**
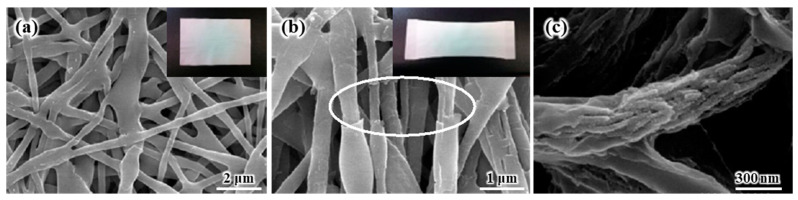
SEM and photo images of the composite membranes after (**a**) heat-pressing at 120 °C, followed by (**b**) heat-stretching at 150 °C (stretching ratio: 71.4%), and (**c**) after removing the PVdF component from the as-electrospun SiO_2_/PVdF composite nanofiber by extraction using acetone for one week.

**Figure 9 polymers-16-01810-f009:**
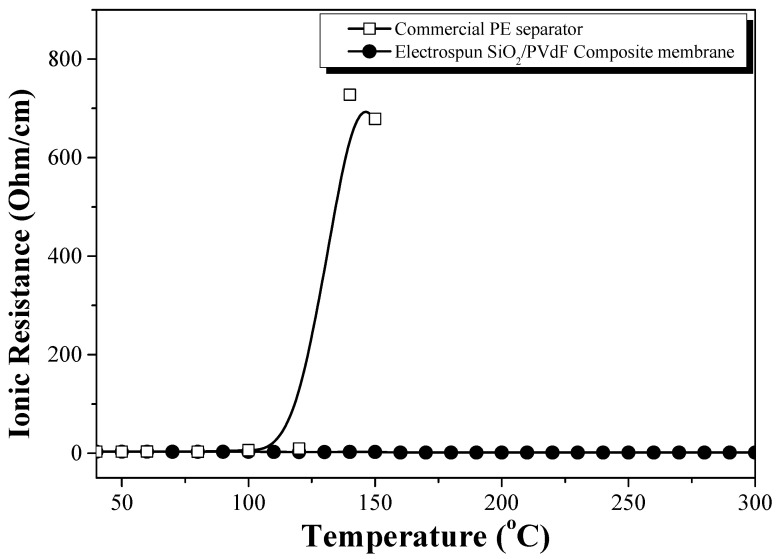
Temperature dependence of the ionic resistivity in the SUS/EPEM/SUS cells. (Reference: commercial PE separator with a similar porosity).

**Figure 10 polymers-16-01810-f010:**
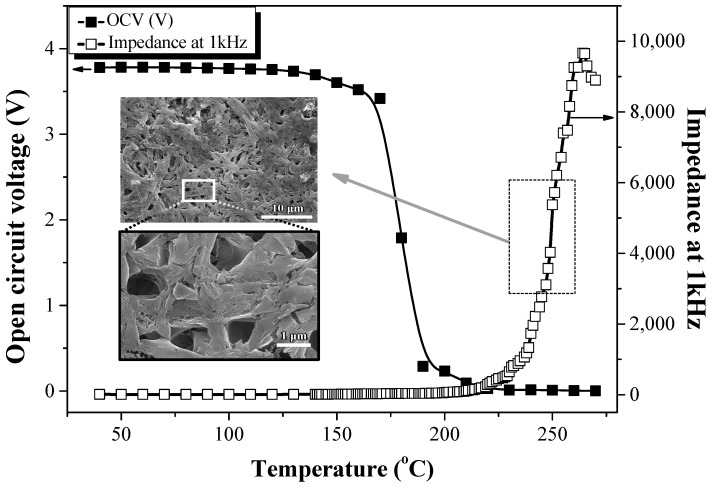
Thermal stability of the prototype cell comprising the electrospun SiO_2_/PVdF blend nanofiber-based composite membrane containing 1 M LiPF_6_–EC/PC/DEC/VC. (Inset image: SiO_2_/PVdF blend nanofiber-based composite membrane after OCV and AC impedance measurements up to 270 °C).

**Figure 11 polymers-16-01810-f011:**
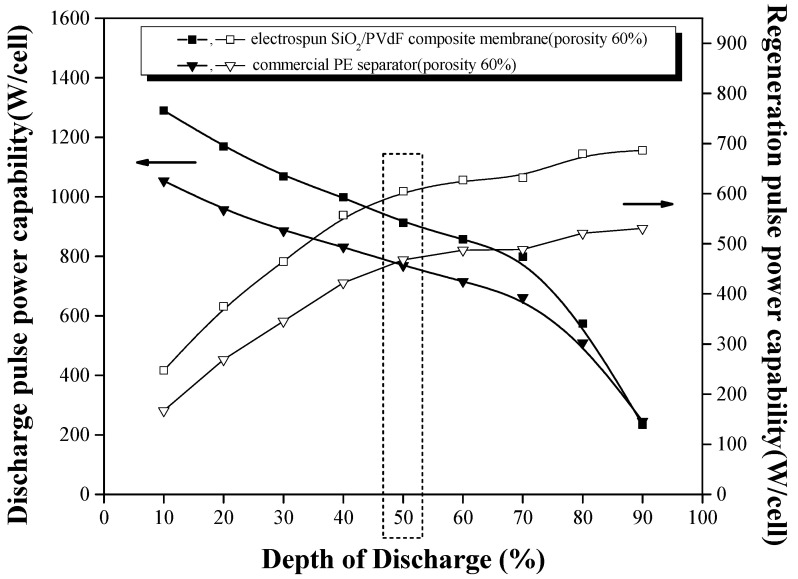
HPPC test of the electrospun SiO_2_/PVdF blend nanofiber-based composite membrane in 1 M LiPF_6_–EC/PC/DEC/VC (reference: commercial PE separator with similar porosity).

**Figure 12 polymers-16-01810-f012:**
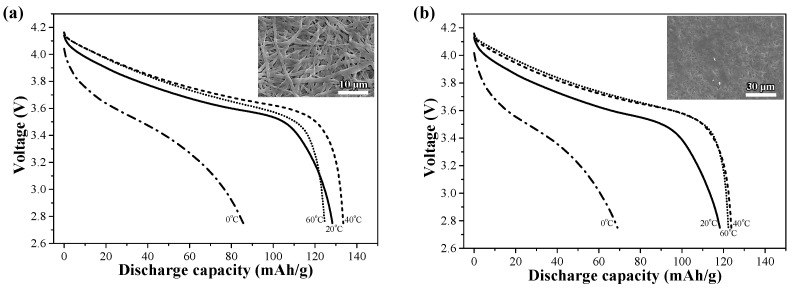
Temperature dependence of the discharge capacity of the prototype cell with EPEM. Porosity values of (**a**) ~60% and (**b**) 40% obtained from the posttreatments, including heat-pressing and heat-stretching.

**Table 1 polymers-16-01810-t001:** Mechanical properties of the electrospun SiO_2_/PVdF (3:7 wt. ratio) blend membrane after heat-pressing and heat-stretching.

	Porosity (%)	Membrane Thickness (μm)	Tensile Stress (MPa)	Tensile Strain (%)	Modulus (MPa)
As-spun membrane ^(1)^ (average fiber diameter ~310 nm)	86~89	85.0	4.3	30.4	74.5
After heat-pressing (120 °C)	~60	27.0	8.8	20.1	138.8
After heat-pressing (120 °C) and heat-stretching (71.4% ^(2)^ at 150 °C)	-	30.0	26.5	13.3	348.0

(1) As-spun membrane was prepared under an applied voltage of 10.5 kV, a tip-to-grounded collector distance of 14 cm, a needle size of 30 G, and a solution feeding rate of 30 μL/min. (2) Membrane was broken under heat-stretching of 84.2%.

## Data Availability

The original contributions presented in the study are included in the article/Appendix A, further inquiries can be directed to the corresponding authors.

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
