# Peer review of "Electrospun Silicon Dioxide/poly(vinylidene fluoride) Nanofibrous Membrane Comprising a Skin Multicore–Shell Nanostructure as a New High-Heat-Resistant Separator for Lithium-Ion Polymer Batteries"

_polymers, 2024, doi:10.3390/polym16131810_

Round 1
Reviewer 1 Report
Comments and Suggestions for Authors
This paper reports electrospun SiO2/PVdF blend fiber-based composite membrane and its application in LIB. This paper can be published after major revison.
1. Can the presence of SiO2 in the fibers be characterized by TEM, XRD?
2. What is the content of SiO2?
3. How does the SiO2 skin form?
4. Figure 5 displays abnormally.
5. The format of Figure. 6
6. In Figure 8b, the temperature does not accurately correspond to the line.
7. Figures are too scattered.
8. The format of the reference
9. Why is the conclusion divided into two paragraphs
Comments on the Quality of English Language
This paper needs to be polished further.
Author Response
Thank you for your Comments and Suggestions
According to the reviewers comments, the manuscript was revised as follows,
- References were corrected to the “polymers journal formation” and some references were exchanged to new papers. Also, the parts we modified are highlighted in grey.
- To cope with the aims and scope of the “polymers” journal, the new data and some explanations were provided in the revised manuscript.
New figures of Fig.1~7 were added to clearly conform the skin multicore-shell nanostructure and its formation mechanism of electrospun SiO2/PVdF blend fiber. And we also provided new data (Figure S2~S4) in the Supplementary Data for them.
- The battery performance data in view of just thermal stability of the skin multicore-shell composite were still retained, but some data such as Figure 6(the electrochemical stability window), Figure 9(C-rate performance), and Figure 10(cycle performance) were deleted.
- Every highlighted parts according to the reviewers advices were corrected as shown in author’s opinions for comments and questions of each reviewer.

Reviewer 2 Report
Comments and Suggestions for Authors
The manuscript “polymers-3007484” reports preparation and characterization of high-temperature resistant separator for Li-ion batteries. In my opinion the paper is important and interesting but it hardly falls into aims and scope of the “polymers” journal since it addresses performance of the cell with new separator but does not give an insight in aspects of polymer science. This may be changed if a detailed answer on question 9 of the following review would be given in the paper. Alternatively, the paper would be much more suitable in other MDPI journal such as membranes, materials, or energies.
1. As far as I know, there is no lithium metal in Li-ion batteries, so the sentence in introduction about ignition temperature should be rewritten. The authors probably meant ignition of LiCoO2.
2. The text in the second paragraph of the 2.4. section seem to repeat what was already mentioned in the fourth paragraph of the 2.3. section.
3. In the end of page 9 the authors mention heat-pressing and heat-stretching of the membranes. But I could not find detailed information on these processes in experimental part. Also the authors explained reasons to choose pressing temperature of 120°C but did not do that regarding stretching temperature. Later, calcination is mentioned but it also was not described in detail.
4. Why, according to Table 1, the membrane thickness increased after stretching? I would expect the opposite.
5. Figure 5 is unreadable in PDF version of the paper since the bottom axis is not shown.
6. I wonder if there is a missing decimal separator in value of electrolyte uptake of 831% in page 18.
7. Why there are local minimums and maximums in dependency of the prototype cell discharge capacity on the cycle number?
8. Are there any industrial or so processes that require operation of Li-ion battery at temperatures higher than 150°C? If yes, please add this information to introduction. I understand that it is not worse to have the membrane working even at such high temperature, but why is it better?
9. Why in authors opinion in the process of phase separation silica tends to form shell and continuous core fibers instead of being randomly dispersed across the PVDF matrix as droplets etc.?
Comments on the Quality of English LanguageThere are some typos in the paper text: viscose – viscous; bellows – described in; proptylene – propylene;
Author Response

(The authors gave the same response as above.)

Round 2
Reviewer 1 Report
Comments and Suggestions for Authors
this paper is revised well.
Author Response
Thank you for kind review and comment.
Reviewer 2 Report
Comments and Suggestions for Authors
The paper was reworked almost from scratch and in my opnion it is now much more suitable for the polymers journal. The so-called multicore shell structure is now discussed in detail and mechanism of ots formation is explained.
The revised version of the paper in my opinion can be accepted.
I have only a few optional recommendations for the authors:
1. If I understood it correctly the authors prepared samples of composite membranes using fumed silica and such membranes did not posess multicore shell structure. However I could not find information on the characteristics of fumed silica used. So consider adding a separate section "materials" with characterisitcs of all the materials used.
2. In my opinion it would be advantageous to add a table with sample codes and parameters of the membrane formation process. For me it was still a bit hard to keep track of different samples without such table. The postprocessing applied and its parameters could also be noted in this table.
Comments on the Quality of English LanguageThere are some typos in the paper text, for example, electrospOn in Line 119.
Author Response
1. If I understood it correctly the authors prepared samples of composite membranes using fumed silica and such membranes did not posess multicore shell structure. However I could not find information on the characteristics of fumed silica used. So consider adding a separate section "materials" with characterisitcs of all the materials used.
Answer : Section 2.1 data has been added, and the revised parts are highlighted with yellow lines.
2. In my opinion it would be advantageous to add a table with sample codes and parameters of the membrane formation process. For me it was still a bit hard to keep track of different samples without such table. The postprocessing applied and its parameters could also be noted in this table.
Answer : Our analyses predominantly featured electrospun SiO2/PVdF (3:7 wt. ratio) samples, with a few exceptions (Figures 3 and 5) as specified in the respective titles. It's important to note that the electrospinning conditions, regardless of the ratio, were consistently maintained and are thoroughly explained in Section 2.2. Therefore, we have tabulated the mechanical properties of the SiO2/PVdF (3:7 wt. ratio) sample and supplemented additional content.